# The Small GTPase RAC1B: A Potent Negative Regulator of-and Useful Tool to Study-TGFβ Signaling

**DOI:** 10.3390/cancers12113475

**Published:** 2020-11-22

**Authors:** Hendrik Ungefroren, Ulrich F. Wellner, Tobias Keck, Hendrik Lehnert, Jens-Uwe Marquardt

**Affiliations:** 1First Department of Medicine, Campus Lübeck, University Hospital Schleswig-Holstein, D-23538 Lübeck, Germany; Jens.Marquardt@uksh.de; 2Clinic for General Surgery, Visceral, Thoracic, Transplantation and Pediatric Surgery, Campus Kiel, University Hospital Schleswig-Holstein, D-24105 Kiel, Germany; 3Clinic for Surgery, Campus Lübeck, University Hospital Schleswig-Holstein, D-23538 Lübeck, Germany; ulrich.wellner@uksh.de (U.F.W.); tobias.keck@uksh.de (T.K.); 4University of Salzburg, A-5020 Salzburg, Austria; hendrik.lehnert@sbg.ac.at

**Keywords:** pancreatic ductal adenocarcinoma, RAC1, RAC1B, transforming growth factor β, cell migration, signaling, cancer, Smad

## Abstract

**Simple Summary:**

Transforming growth factor β (TGFβ) promotes pancreatic ductal adenocarcinoma (PDAC) primarily through its non-canonical (non-Smad) signaling arms, including signaling by the small GTPase RAC1. The human RAC1 gene also encodes for another protein, designated RAC1B, but whether this isoform also interacts with TGFβ signaling has remained unknown. In a series of studies in PDAC-derived cells, we found that RAC1B also cross-talks with TGFβ signaling, but unlike RAC1 antagonizes TGFβ-induced responses, i.e., epithelial–mesenchymal transition, through multiple mechanisms. However, rather than being uniformly inhibitory, RAC1B selectively blocks tumor-promoting pathways, while concomitantly allowing tumor-suppressive pathways to proceed. In this review article, we discuss the specific interactions between RAC1B and TGFβ signaling, which occur at multiple levels and include various components of both the canonical Smad and non-Smad pathways. In addition to emerging as a novel tumor suppressor in PDAC, RAC1B turned out to be a useful tool to dissect TGFβ signaling.

**Abstract:**

RAC1 and its alternatively spliced isoform, RAC1B, are members of the Rho family of GTPases. Both isoforms are involved in the regulation of actin cytoskeleton remodeling, cell motility, cell proliferation, and epithelial–mesenchymal transition (EMT). Compared to RAC1, RAC1B exhibits a number of distinctive features with respect to tissue distribution, downstream signaling and a role in disease conditions like inflammation and cancer. The subcellular locations and interaction partners of RAC1 and RAC1B vary depending on their activation state, which makes RAC1 and RAC1B ideal candidates to establish cross-talk with cancer-associated signaling pathways—for instance, interactions with signaling by transforming growth factor β (TGFβ), a known tumor promoter. Although RAC1 has been found to promote TGFβ-driven tumor progression, recent observations in pancreatic carcinoma cells surprisingly revealed that RAC1B confers anti-oncogenic properties, i.e., through inhibiting TGFβ-induced EMT. Since then, an unexpected array of mechanisms through which RAC1B cross-talks with TGFβ signaling has been demonstrated. However, rather than being uniformly inhibitory, RAC1B interacts with TGFβ signaling in a way that results in the selective blockade of tumor-promoting pathways, while concomitantly allowing tumor-suppressive pathways to proceed. In this review article, we are going to discuss the specific interactions between RAC1B and TGFβ signaling, which occur at multiple levels and include various components such as ligands, receptors, cytosolic mediators, transcription factors, and extracellular inhibitors of TGFβ ligands.

## 1. Introduction

The Rho family of small GTPases includes twenty members which are involved in the regulation of actin cytoskeleton remodeling, cell polarity, adhesion, and migration, and which are also involved in cell proliferation, differentiation and stem cell maintenance. One of the best-characterized members of this family, with a ubiquitous expression in diverse tissues, is RAC1 [1,2]. The activity of RAC1 is regulated by three types of accessory proteins. Guanine nucleotide exchange factors (GEFs) favor the exchange of GDP with GTP, whereas GTPase-activating proteins (GAPs) accelerate the GTP/GDP conversion and hence the on/off switch. The guanosine nucleotide dissociation inhibitor (GDI) binds to the GDP-bound forms, prevents the GDP/GTP exchange, and hence arrests the protein in the off-state, but also sequesters RAC1 in the cytoplasm [3]. RAC proteins are also regulated by posttranslational modifications, such as prenylation of the C-terminal CAAX motif favoring membrane interaction, phosphorylation, SUMOylation, and ubiquitination [3] (Figure 1). RAC1 can function on its own or assemble with other signaling proteins to form multi-subunit catalytic complexes, i.e., the NAD(P)H oxidase (NOX) complex, which generates reactive oxygen species (ROS) [2,3]. Together, this emphasizes the importance of fine-tuning the activity, subcellular targeting, and eventually the higher-order assembly of RAC to allow for the selective activation of signaling cascades and cellular responses.

RAC1 has multiple downstream effectors and initiates several signaling pathways, including p21-activated kinases (PAKs), NOX1, NFκB, mitogen-activated protein kinases (MAPKs), and Wnt/β-catenin to regulate membrane ruffling, lamellipodia formation, and cell–cell contacts, as well as cell polarity, adhesion, and motility [3]. Since these changes impact cell proliferation, epithelial–mesenchymal transition (EMT) and invasiveness, it is not surprising that RAC1 is involved in neoplastic transformation and cancer progression by means of its ability to promote stemness features as well as invasion, metastatic dissemination, and even drug- and radioresistance [4,5]. RAC1-driven cancer cell invasion involves both single-cell and collective modes of migration. Single-cell migration is the predominant type of cell movement in poorly differentiated, mesenchymal tumors [6], whereas collective migration is more representative of differentiated epithelial tumors [7,8].

In mammals, the *RAC1* gene can give rise to an alternatively spliced isoform, termed RAC1B (Figure 1). RAC1B has a more restricted tissue distribution and is expressed in a cell type- and differentiation-dependent (and presumably also disease-associated) context. The RAC1B protein is also abundantly expressed in chronic pancreatitis [9] and in human inflammatory colonic mucosa [10], and its upregulation in intestinal epithelial cells of transgenic mice contributes to intestinal wound healing after acute inflammation [11]. RAC1B mRNA or protein has also been detected in various human cancers, including those of the breast, thyroid, ovary, colon, pancreas, and lung [3]. Furthermore, among panels of established adenocarcinoma cell lines of pancreatic, lung, and breast origin, RAC1B protein expression was found to be associated with a differentiated, epithelial histomorphologic phenotype [12,13] or—according to another classification—a partial EMT phenotype [14]. In contrast, its protein and transcript levels were low or absent in (quasi)mesenchymal or basal-like cell lines [12,13] predicted to have a complete EMT phenotype [14]. The generation of the RAC1B variant is the result of the inclusion of an additional exon (exon 3b) of *RAC1*, which leads to a 19-amino acid in-frame insertion immediately C-terminal to the switch II domain (Figure 1). The ratio of RAC1 to RAC1B is regulated by splicing factors such as SRSF3 (formerly SRp20) and SRSF1 (formerly ASF1/SF2) that promote either the skipping or the inclusion, respectively, of exon 3b [15,16]. For instance, the Wnt pathway favors the excision of exon 3b via the induction of SRSF3 [15] (Figure 1), whereas epidermal growth factor receptor (EGFR) favors its inclusion by activation of SRSF1 [16].

When compared to RAC1, RAC1B exhibits a number of distinctive features. It preferentially exists in a GTP-bound and hence active form as a result of its reduced intrinsic GTPase activity and impaired binding to Rho-GDI [17,18]. However, RAC1B may still be activated further by extrinsically-activated GEFs. A comparison of the activation states of RAC1B in benign and malignant pancreatic epithelial cell lines showed that the amount of active RAC1B in a given cell is mainly regulated by expression of the protein rather than by modulation of its activity [12]. These observations suggest that the abundance of the protein in a given tissue grossly reflects its biological activity. Unlike RAC1, RAC1B does not activate PAK1, AKT1, JNK, or the transactivation activity of RelB-NFκB2/p100 [18,19,20]. RAC1B’s failure to stimulate PAK1 may underlie its inability to induce lamellipodia formation and to disrupt cell–cell contacts in keratinocytes [21]. The extra 19-amino acid sequence, however, endows RAC1B with novel functions as a consequence of enhanced binding to proteins involved in cell–cell adhesion, motility, and transcriptional regulation such as SmgGDS, RACK1, and p120 catenin [22]. RAC1B has also been reported to interact with Dishevelled-3 and to form a tetramer with β-catenin/TCF that is recruited to the promoter of canonical Wnt target genes [23]. As a consequence of impaired binding to Rho-GDI, RAC1B is primarily localized at the inner leaflet of the plasma membrane, which brings it into contact with specific interaction partners or substrates, especially membrane receptors and their signalosomes, i.e., those for transforming growth factor β (TGFβ).

As a result of their multiple interactions with other proteins, RAC1 and RAC1B play a central role in mediating cross-talk of different cellular signaling networks. Their strategic positions within the cell allows them to transmit signals from membrane receptors and activate downstream pathways that eventually result in changes in gene expression and complex cellular responses. Depending on their activation state, they shuttle between different subcellular locations and their translocation to the inner cell membrane or even to the nucleus further increases the spectrum of potential interaction partners and hence functional activities. This makes RAC1 and RAC1B ideal candidates to establish cross-talk between two or more signaling pathways. Cancer cells may exploit this to increase their aggressiveness, i.e., by selectively modulating the activity of pro-invasive/pro-metastatic pathways. A prototype example in this respect is the interaction of RAC1/RAC1B with signaling induced by TGFβ, which we shall discuss in this article.

## 2. Basics of TGFβ Signaling

To better understand the various mechanistic interactions of RAC1/RAC1B with TGFβ signaling to be discussed in detail later, some fundamentals of TGFβ biology will be introduced first. The mammalian genome encodes three TGFβ proteins, termed TGFβ 1, 2, and 3. Each gene encodes a precursor protein with a short amino-terminal signal peptide and a carboxy-terminal mature TGFβ polypeptide of 112 amino acids linked by a long pro-segment. The mature TGFβ dimer is secreted in a “latent”, biologically inactive form called “latency-associated polypeptide” (LAP) that prevents TGFβ binding to its cell surface receptors [24]. Subsequent activation is required to release active TGFβ that eventually binds—as a disulfide-linked dimer—to TGFβ receptors on cells in close proximity [24,25].

Active TGFβ binds in a tetrameric complex to two types of transmembrane serine/threonine kinases, designated type I (TβRI) and type II (TβRII) receptors [26]. TGFβ binding to these receptor complexes activates the receptors and consequently the intracellular signal transducers SMAD2 and SMAD3 through C-terminal serine phosphorylation by the type I receptor kinase, also designated activin receptor-like kinase 5 (ALK5) (Figure 2). These receptor-regulated Smads combine with the common-mediator Smad, SMAD4 (encoded by the tumor suppressor gene *DPC4*), and the resulting complexes are translocated to the nucleus. Here, they bind to regulatory gene sequences, e.g., Smad-binding elements in promoters of TGFβ target genes, or associate with DNA sequence-specific transcription factors and coregulators to activate or repress gene transcription in response to TGFβ [26] (Figure 2). Following ligand binding, the complex is internalized and ALK5 is eventually degraded by the inhibitory Smad, SMAD7 (Figure 2). SMAD7 itself is a TGFβ target gene and is rapidly transcriptionally induced by Smad signaling in response to ligand stimulation to provide negative feedback inhibition to attenuate Smad signaling [27]. SMAD7 is also regulated at the level of protein stability by ubiquitination and deubiquitination [27]. Recently, the deubiquinating enzyme ubiquitin-specific-peptidase 26 (USP26) has been shown to target SMAD7 and to be a potent negative regulator of TGFβ signaling [28].

Although Smad signaling is referred to as “canonical” signaling for TGFβ family proteins, TGFβ can also activate non-canonical Smad or non-Smad signaling in epithelial cells. Non-canonical Smad signaling occurs through BMP-type Smads SMAD1 and SMAD5 in response to TGFβ—although with a more transient kinetics—as a result of a cooperation between ALK5 and ALK2 (encoded by *AVCR1*, a type I receptor with BMP-like Smad1/5 signaling) [29]. Of note, at the transcriptome level almost a quarter of the TGFβ activated genes in a murine mammary epithelial cell line were found to depend on SMAD1/5 in addition to SMAD3 [29]. Non-Smad signaling pathways that are usually more strongly activated by receptor tyrosine kinases include the ERK and p38 MAPKs, PI3K/AKT pathways, and RAC/RHO signaling [26,30] (Figure 2).

The number of TGFβ receptors at the cell surface of non-stimulated cells is low when compared to receptor tyrosine kinases, because the bulk of TGFβ receptors are retained intracellularly and are not available for TGFβ binding [26]. Newly synthesized TβRI and TβRII are delivered from the Golgi apparatus to the cell surface via separate routes. Both ligand-occupied and unoccupied TGFβ receptors at the cell surface are distributed between different microdomains of the plasma membrane and are constantly internalized by two routes of endocytosis, clathrin- and caveolae-mediated endocytosis (Figure 2). Although receptor endocytosis is not essential for TGFβ signaling, it plays a direct role in determining the type of signaling pathway(s) activated. Localization of the activated receptor complexes on the early endosomes, endocytosed via clathrin-mediated endocytosis, promotes TGFβ-induced Smad activation and signaling, whereas receptor endocytosis via clathrin-independent, membrane raft/caveolae-mediated endocytosis is required for TGFβ signaling via non-Smad pathways, in addition to facilitating the degradation of TGFβ receptors [31] (Figure 2). TβRIII, a non-transducing accessory receptor, has been identified as a critical regulator of clathrin-mediated endocytosis of type I and type II receptors via signaling to early endosome antigen 1 (EEA1) and Rab5-positive early endosomes [32,33]. Since early endosome signaling is controlled by Rab5 [32,33] and Rab5 may be required for the activation of RAC1 [34], it is possible that both anterograde and retrograde transport of TGFβ receptors and hence their steady-state cell surface levels are regulated by RAC1 or RAC1B.

The availability of the receptors on the cell surface is controlled by several mechanisms. Stimulation with ligand induces rapid anterograde transport of receptors to the cell surface, thus enhancing their availability for TGFβ binding and consequently enhancing TGFβ responsiveness [35]. However, while short pulses of ligand exposure increase TGFβ responsiveness, extended or chronic exposure of cells to autocrine or paracrine TGFβ eventually results in desensitization, a period during which the cell becomes refractory to further stimulation with ligand [36].

## 3. Deregulation of TGFβ Signaling in Cancer Cells

TGFβ signaling is known to be regulated in a cell type-, context-, and spatiotemporal-specific manner. Not surprisingly, therefore, its deregulation can cause severe pathologies such as fibrosis/desmoplasia and cancer. Cancer-relevant cellular processes that are controlled by TGFβ include growth inhibition, epithelial–mesenchymal transition (EMT), and EMT-associated events, i.e., invasion/metastasis, (cancer) stem cell generation, as well as drug resistance. Both cell-intrinsic signals and clues from the TME shape the final TGFβ response, which suggests extensive cross-talk between canonical TGFβ and non-TGFβ signaling and even among the canonical and non-canonical arms of the TGFβ signaling machinery. TGFβ/Smad signaling also induces extensive changes in mRNA splicing, thus generating different protein isoforms [37] through its ability to control the expression of SRSF3 [38]. Since this splicing factor also negatively regulates the generation of RAC1B (Figure 1), TGFβ may be able to directly decrease the ratio of RAC1B to RAC1 expression, thereby facilitating mesenchymal differentiation.

TGFβ signaling in cancer cells is regulated by both genetic and epigenetic alterations. Mutational inactivation of *DPC4* is a frequent and late event in the progression of PDAC, also detected in other gastrointestinal tract carcinomas, as well as in some extra-gastrointestinal carcinomas [39]. Mutations in *DPC4* do not necessarily inactivate TGFβ/Smad signaling [40,41]. Rather, in genetically engineered mouse models of PDAC, heterozygous mutations of *Dpc4* in the cancer cells attenuate their metastatic potential, while increasing their proliferation [42]. Moreover, in human patient-derived organoids the mutational status of *DPC4* has been shown to determine the mode of cell migration. Organoids from cancers with somatic mutations in *DPC4* exhibited collective invasion but nevertheless required exogenous TGFβ. Interestingly, RAC1 and CDC42 were identified as potential mediators of TGFβ-stimulated collective invasion in the *DPC4*-mutant PDAC organoids [8].

Although genetic mutations involved in regulating activity of TGFβ signaling have been fairly well characterized, this not the case for epigenetic mechanisms. In advanced tumors, the *TGFBR2* or *SMAD2* genes are frequently silenced by histone and DNA methylation [43]. As mentioned above, increased expression of SMAD7 in various carcinomas attenuates Smad signaling and eventually degrades ALK5 (Figure 2), whereas increased expression of the Smad corepressors c-Ski or SnoN selectively impairs Smad-mediated transcription [27]. Negative control of canonical TGFβ signaling through cross-talk with non-TGFβ or non-canonical TGFβ pathways has also been demonstrated. For instance, oncogenic RAS via ERK MAPKs can inhibit TGFβ signaling in mammary and lung epithelial cells by blocking nuclear accumulation of SMADs 2 and 3, and hence Smad-dependent transcription, through phosphorylation events in their linker regions [44]. These examples reveal a remarkable feature of the TGFβ pathway: extensive control by negative feedback inhibition. As discussed later, RAC1B may be another crucial player here.

Resting epithelial cells show very low, if any, expression of TGFβs in vivo. Hyperplasia and neoplasia confer the ability for their increased synthesis and secretion [26], followed by growth suppression of benign or low-grade cancers, whereas in their malignant and metastatic counterparts the increased production of TGFβ has been reported to stimulate cancer progression in an autocrine or paracrine manner. The increased sensitivity of most carcinoma cells when compared with normal epithelial cells to autocrine or localized TGFβ signaling [26] and the ability of TGFβ1 to stimulate its own synthesis or increase the number of its receptors on the cell surface [35] may eventually result in a feed-forward loop and a vicious cycle of the TGFβ response [45]. However, as discussed above, autocrine TGFβ may not always enhance TGFβ responses but in specific contexts may be inhibitory, i.e., by desensitizing cancer cells towards the action of non-autocrine TGFβ. As such, autocrine TGFβ may behave as a tumor suppressor through its ability to block the protumorigenic action of stromal, matrix-derived TGFβ (see below).

## 4. Interaction of RAC1 with TGFβ Signaling

In both benign and malignant cells, the cross-talk of RAC1 with TGFβ signaling promotes a diverse array of cellular responses. In several cell types, including PDAC-derived cells [46,47,48], RAC1 is activated by TGFβ1 and modulates protein levels, i.e., of cyclins and p27^Kip1^ [47], connective tissue growth factor [49], type I collagen [50], and MMP9, as well as complex cellular responses such as epithelial cell plasticity [51] and endothelial morphogenesis [52]. In addition, TGFβ stimulates RAC1 activity in HER2-overexpressing MCF10A human mammary epithelial cells (HMECs) cells and its association with HER2 [53]. Conversely, RAC1 can regulate TGFβ signaling, i.e., mutant HER2 in MCF10A cells activates autocrine TGFβ1 signaling through a mechanism involving RAC1-dependent transcription, and MCF10A cells transformed by oncogenic H-RAS^G12V^ also express higher TGFβ1 levels through RAC1 activation [54]. RAC1 also increases the cigarette smoke-induced TGFβ release and EMT [55], while RAB23-mediated RAC1 activation and subsequent TGFβ expression drives HCC migration [56]. RAC1 and its ability to produce ROS has been shown to be required for efficient TGFβ1-induced activation of SMAD2 and p38 MAPK in keratinocytes and PDAC cells [48,57,58] and in the latter type of cells both events are required for adherence-dependent TGFβ induction of BGN [48]. BGN is overexpressed in fibrotic tissues in accordance with RAC1 promoting fibrotic TGFβ1 signaling and chronic kidney disease [59]. Moreover, RAC1 is an upstream activator of MEK-ERK, PI3K/AKT, and PAK2 signaling, all of which can be activated by TGFβ in a cell type-specific manner [60]. RAC1 is also involved in TGFβ control of cell spreading [61], EMT, migration, invasion, and metastasis. RAC1, via PAK1, drives TGFβ1-induced prostate cancer cell EMT [62], as well as EMT and tumor growth in gastric cancer [63]. TGFβ1 signaling to PI3Kγ/AKT/RAC1 or through RAC1/ROS mediates cell migration of hepatic stellate cells in liver fibrosis [64] or invasiveness of PDAC cells, respectively [65,66], whereas TGFβ/Smad3 signaling through DOCK4 and RAC1 facilitates lung adenocarcinoma metastasis [67].

Of note, all these studies reported positive interactions between RAC1 and TGFβ signaling and there is no evidence for the existence of a second RAC1 isoform that acts here as an endogenous inhibitor or antagonist of RAC1. This situation, along with the fact that RAC1B is expressed more variably and at considerably lower levels than RAC1 [18] and, in addition, differs from RAC1 by altered downstream signaling (see above), led us to believe that RAC1B’s functions are distinct from—or even antagonistic to—those of RAC1. Indeed, there are examples from other genes where the alternatively spliced product functionally antagonizes the canonical or parental form [68,69], i.e., MiniSox9 and Sox9 [70], Bcl-X_S_ and Bcl-X_L_ [71], and N-terminal truncated DNp73 isoforms and TAp73 [72]. With respect to BCL-X during malignant progression, generation of the short, pro-apoptotic, and thus anti-oncogenic BCL-X_S_ variant (resulting from alternative splicing and lacking 63 amino acids of BCL-X_L_) is silenced to unleash the oncogenic potential of the long, anti-apoptotic, and thus pro-oncogenic Bcl-X_L_ isoform. This resembles the situation with RAC1 and RAC1B in PDAC, where RAC1B levels were higher in cell lines established from well-differentiated, epithelial-like tumors with lower invasive abilities than in cell lines retrieved from poorly differentiated, mesenchymal-like tumors with high invasive potential [12,13], suggesting that the expression of RAC1B was lost during progression towards a more aggressive phenotype. The overall effect of the tumor suppressor protein p73 in oncogenesis is thought to depend on the ratio of the TAp73 to DNp73 isoforms [72] and a similar scenario may hold for RAC1 and RAC1B. Indeed, while studying the cross-talk of RAC1B and RAC1 with TGFβ signaling in pancreatic and breast epithelial cells, we realized that RAC1B, in contrast to RAC1, inhibited rather than promoted some complex responses to TGFβ such as growth inhibition [73], EMT [74], and cell migration [9,65,73,74]. Based on the antagonistic behavior of RAC1 and RAC1B, which extended to the regulation of individual genes in some cases, the specific function of RAC1B could eventually be deduced from that of RAC1.

In a series of publications, we have revealed an unexpected array of mechanisms through which RAC1B cross-talks with TGFβ signaling. In the Section 4.1 through Section 5 we are going to discuss the specific interactions and mechanisms of RAC1B with TGFβ pathway components that encompass all levels of the cascade, such as ligands, receptors, cytosolic signaling mediators, transcription factors, and extracellular inhibitors of TGFβ ligands. A diagram summarizing both the already published and still hypothetical interactions is presented in Figure 3.

### 4.1. Receptors

#### 4.1.1. The TGFβ Type I Receptor ALK5

The TGFβ type I receptor, ALK5, is a central regulator and bottleneck in TGFβ signaling since it is the only signal transducing receptor for TGFβ in most cell types, except for ALK1 in endothelial cells [26]. Although other type I receptors may aid in transducing the TGFβ signal, i.e., ALK2 in activation of SMAD1/5 [29], these depend on ALK5 function [29]. From an evolutionary perspective it is therefore not surprising that ALK5 is subject to complex positive and negative control at multiple levels, e.g., de novo transcription, control by microRNAs [75], posttranslational modification by core fucosylation [76], internalization/endocytosis [31], and protein stability by ubiquitination and deubiquitination [77]. Intriguingly, we found that RAC1B interferes with some of these processes to efficiently target ALK5 for inhibition:(i)RAC1B inhibits autoinduction of TGFBR1 mRNA by TGFβ1. This became apparent only after knock-down or knock-out of RAC1B when ALK5 mRNA abundance increased in a time-dependent manner in response to stimulation with recombinant human TGFβ1 (rhTGFβ1) [73]. These data are compatible with *TGFBR1* being a TFβ response gene [26,35].(ii)RAC1B inhibits proteins required to sustain ALK5 protein expression. We have previously shown that proteinase-activated receptor 2 (PAR2) encoded by *F2RL1* was required for TGFβ1 signaling by its ability to sustain protein expression of ALK5 by an as-yet-unknown mechanism [78]. The combined knock-down in the PDAC cell line, Panc1, of RAC1B and PAR2 relieved the stimulatory effect of RAC1B single knock-down on the abundance of endogenous ALK5 mRNA and migratory activity, indicating that PAR2 is involved in mediating the suppressive effect of RAC1B on ALK5 and ALK5-dependent cell migration. Conversely, *F2RL1* itself is a TGFβ target gene [79] and its mRNA expression was induced by treatment of cells with rhTGFβ. Consequently, the RAC1B knock-down-induced rise in PAR2 mRNA was likely the result of derepressed ALK5 levels activated by autocrine TGFβ, since PAR2 upregulation was relieved by co-transfection of RAC1B small interfering RNA (siRNA) with ALK5 siRNA [79]. This indicated that RAC1B disrupts an autoregulatory feed-forward loop between ALK5 and PAR2.

Recently, a novel regulatory role for PAR2 in the anterograde traffic of the related PAR4 has been described. Specifically, PAR2 facilitated plasma membrane delivery of PAR4 and also enhanced its glycosylation and activation of signaling [80]. The TGFβ receptors share characteristics with PAR4, i.e., poor cell surface expression, the retainment in the endoplasmatic reticulum (ER), and the requirement of N-linked glycosylation or core fucosylation for their successful localization at the cell surface [81]. Moreover, folding and processing of TβRI/ALK5 in the ER has been shown to be inefficient and could thus serve as a mechanism for controlling the number of plasma membrane receptors [31]. PAR2 is a potential candidate here that might function as a chaperone for ALK5 in its anterograde transport to the cell surface. This would be compatible with earlier results showing that PAR2 protein—but not its proteolytic activation or signaling function [82]—was required for sustaining ALK5 expression in pancreatic epithelial [78] and hepatic stellate cells [83]. Hence, inhibition of PAR2 would enable RAC1B to efficiently prevent ALK5 from reaching the cell surface and thus keep the cells’ sensitivity to TGFβ low.

(iii)RAC1B promotes ALK5 protein degradation via induction of SMAD7. Activated ALK5 is known to recruit SMAD7 in a complex with the E3 ligase Smurf2 to promote its internalization and eventual proteasomal degradation in order to terminate TGFβ signaling. Earlier, we demonstrated that the suppressive effect of RAC1B on ALK5 was dependent on SMAD7, and further that RAC1B upregulated protein expression of SMAD7 via intermittent induction of USP26 (see Section 4.2.1) [84].

#### 4.1.2. Other Type I Receptors Involved in TGFβ Signaling

TGFβ is also capable of BMP-type signaling through ALK5-mediated activation of ALK2, as outlined above [29]. Due to its potent inhibition of ALK5, RAC1B also likely affects ALK5-dependent ALK2 signaling. However, it will be interesting to see if RAC1B targets the ALK2-Smad1/5 branch directly and separately from ALK5. Preliminary data from an expression screening in Panc1 cells lacking RAC1B revealed that the mRNA abundance of ALK2 was increased [85], pointing to co-repression of ALK5 and ALK2. Currently, there is little information regarding cellular responses mediated by TGFβ-induced SMAD1/5 activation, but in light of the different activation kinetics it is conceivable that the ALK2-SMAD1/5 branch functionally differs from the ALK5-SMAD2/3 branch, or even antagonizes it. In a murine mammary epithelial cell line the ALK2-Smad1/5 arm has been implicated in TGFβ-induced EMT and in the induction of the *Id1* gene [29] (see below). It remains to be seen, however, whether this also applies to human epithelial cells and whether RAC1B is able to target this arm of TGFβ signaling.

It is also conceivable that RAC1B targets BMP ligand/BMP receptor signaling, given the finding that BMPs can signal through the canonical TGFβ-responsive Smad2 and Smad3 by stimulating complex formation between the BMP type I TGFβ superfamily receptors ALK3/6 and ALK5/7 [86]. Consistent with increased BMP-mediated Smad2/3 signaling during cancer progression, BMP signaling through Smad3 occurs preferentially in transformed cells and facilitates cancer cell invasion [86]. This suggests that the ability of RAC1B to accelerate ALK5 degradation also co-inhibits BMP-induced pro-invasive SMAD3 signaling. In addition, BMPs can increase the cells’ sensitivity to TGFβ signaling by upregulating TGFβ receptors at the cell surface [35]. Interestingly, an expression screening of RAC1B-depleted Panc1 cells showed that the mRNA abundance of BMP2 and BMP4 was dramatically increased over that of controls [85]. Since the RAC1B-induced downregulation of BMPs is likely associated with reduced secretion, we envisage a scenario in which RAC1B limits BMP signaling in the TME to further decrease the sensitivity of the tumor cells to autocrine or stromal TGFβ.

### 4.2. RAC1B Differentially Affects the Expression or Function of SMAD and MAPKs

#### 4.2.1. Smad Proteins

Canonical TGFβ signal propagation through SMAD2/3 drives most TGFβ responses, whereas non-canonical Smad signaling proceeds through SMAD1/5 as a result of cooperation between ALK5 and ALK2 [29]. We observed that RAC1B affects Smad signaling in a differential fashion:(i)*SMAD2/3*. Due to suppression of ALK5 and its kinase activity by RAC1B, C-terminal serine phosphorylation of SMAD2 and SMAD3 was concomitantly reduced [9]. However, in Panc1-RAC1B knock-out cells, we surprisingly detected lower levels of C-terminally phosphorylated SMAD3 (pSMAD3C) [73], although we expected the opposite as a result of derepression of ALK5. This was a puzzling observation before we realized that the abundance of total SMAD3 protein was dramatically decreased in RAC1B-depleted cells. Interestingly, in the same cells, SMAD2 expression remained unaffected, revealing that RAC1B selectively promoted SMAD3 expression.

Prompted by our hypothesis that RAC1B was a tumor suppressor, we considered the possibility that RAC1B-driven SMAD3 expression fulfills an as-yet-unknown anti-oncogenic function. Intriguingly, we then discovered that SMAD3 was able to mimic the migration-inhibitory effect of RAC1B in several pancreatic and breast epithelial cell lines. The inhibition of invasion was independent of SMAD3 C-terminal serine phosphorylation [13] and likely depended on SMAD3′s ability to induce—in a TGFβ-independent manner—expression of the invasion suppressor E-cadherin [87] and/or the extracellular TGFβ inhibitor BGN [13]. As outlined below, RAC1B regulation of SMAD3 involves intermittent induction of *TGFB1*.

(ii)*SMAD1/5*. Preliminary data from Panc1 cells indicate that pSMAD1/5C levels in response to TGFβ1 stimulation increase much more strongly in cells in which *RAC1* exon 3b has been deleted [85]. As outlined above, this likely reflects derepression of ALK5 and/or ALK2 and the associated increase in their kinase activities. In the murine cells, Smad1/5, in addition to Smad3 signaling, were also required for TGFβ-induced downregulation of the epithelial marker genes *Cdh1* and *Tjp1*, and upregulation of the mesenchymal marker genes *Acta2* and *Fn1*, together being indicative of EMT [29].(iii)*SMAD4*. In Panc1 cells we observed that following siRNA-mediated knock-down of RAC1B, SMAD4 protein expression was reduced [85]. This suggests that RAC1B coordinately drives the expression of both SMAD3 and SMAD4 in pancreatic epithelial cells. Mechanistically, SMAD4 upregulation may occur through transcriptional activation of *DPC4* or a decrease in SMAD4 ubiquitination [88].(vi)*SMAD7*. In a recent study, we have shown that i) RAC1B promotes the expression of SMAD7 and ii) SMAD7 mediates the suppressive effect of RAC1B on ALK5 protein and its associated kinase activity [84]. We further revealed that upregulation of SMAD7 by RAC1B requires the rapid transcriptional induction of USP26 [84]. The involvement of USP26 strongly suggests that RAC1B increases SMAD7 protein stability by reducing the rate of proteasomal degradation; however, a direct demonstration of RAC1B-induced SMAD7 deubiquitination in pancreatic cells needs experimental verification. Of note, USP26 has been identified as a potent negative regulator of TGFβ signaling in breast cancer and glioma cells [28]. Our demonstration of RAC1B transcriptionally inducing USP26 in Panc1 cells, therefore, provides strong evidence in favor of RAC1B being a TGFβ antagonist in pancreatic epithelial cells. Moreover, since SMAD7 has been implicated in the inhibition of EMT and maintenance of the epithelial phenotype [89], its identification as a RAC1B target gene is a significant observation in light of the strong expression of RAC1B in PDAC cell lines of the epithelial/classical subtype [12,13]. RAC1B’s ability to promote USP26 and SMAD7 expression may thus have a crucial role in inhibiting EMT and promoting mesenchymal–epithelial transition (MET).

#### 4.2.2. MAPKs

Inhibition of MAPK activation by RAC1B is seen in pancreatic Panc1 [12,74] and mammary MDA-MB-231 [90] carcinoma cells. A kinetic analysis of RAC1B-depleted Panc1 cells revealed that RAC1B decreases both the extent and the duration of p38 MAPK activation [74]. RAC1B also suppresses basal and TGFβ1-induced pERK2 but not pERK1 levels [12,74] and suppression of ERK2 activation was crucial for RAC1B to stimulate or repress E-cadherin or vimentin expression, respectively [12]. In the case of ERK inhibition, this is unlikely to be the result of downregulation of ALK5 and its associated tyrosine kinase function [91], since otherwise phosphorylation of both ERK1 and ERK2 should have been affected.

Activation of MEK-ERK signaling mediates oncogene-induced senescence (OIS) in response to oncogenic versions of RAS or its downstream mediator RAF [92,93,94]. Of note, depletion of ERK2, but not ERK1, abrogated oncogenic RAS (HRAS^V12^)-induced senescence in mouse embryonic fibroblasts [94]. The inhibitory effect of RAC1B on ERK2 can possibly explain the antagonistic effect of RAC1B on BRAF^V66E^-induced senescence in colorectal cells, as described earlier [95]. Differential inhibition of ERK2 vs. ERK1 activation by RAC1B and its functional consequences certainly warrants more detailed investigations.

With regard to OIS, it is noteworthy that HMECs in response to oncogenic RAS undergo a p16^INK4a^/RB- and p53-independent form of OIS that requires TGFβ signaling/receptor activation [96]. This matches observations in MCF10A HMECs transformed by oncogenic HRAS^G12V^ or expressing mutant HER2 of higher TGFβ1 production and autocrine signaling, which is activated through a mechanism involving RAC1 activation or RAC1-dependent transcription, respectively [54]. We observed the upregulation of RAC1 and the downregulation of RAC1B, which was associated with increased migratory activity, in finite-lifespan (non-transformed) HMECs during the pre-senescence-to-senescence period [90]. This suggests the exciting possibility of the downregulation of RAC1B or a decrease in the RAC1B:RAC1 ratio caused an increase in ALK5 abundance (see above) that, in turn, promoted the senescent phenotype. Hence, putting these findings together, RAC1B’s ability to antagonize senescence may also operate through the suppression of TGFβ/ALK5 signaling.

There is extensive cross-talk between TGFβ-induced Smad and non-Smad signaling. For instance, ERK can inhibit Smad signaling, i.e., by phosphorylation of SMAD2 or SMAD3 in their linker regions [44]. Hence, by suppressing ERK2 phosphorylation, RAC1B can indirectly promote Smad signaling, thereby compensating for reduced Smad signaling due to RAC1B-mediated degradation of ALK5. Since the related RAC1 is an activator of MEK-ERK signaling, e.g., in ovarian cancer cell EMT [97], the functional antagonism between RAC1 and RAC1B extends to the level of individual MAPK pathways. Since RAC1 also promotes PI3K/AKT signaling and hence cellular responses that are controlled by this pathway, e.g., membrane recruitment of TGFβ receptors in response to high glucose [26] or stimulation with ligand [35], RAC1B may be able to block this process if indeed operating as an endogenous inhibitor of RAC1.

### 4.3. RAC1B Enhances Expression and Secretion of Autocrine TGFβ1: A Possible Role in Tumor Suppression

Many mesenchymal-subtype carcinomas are characterized by high autocrine TGFβ1 production, but the underlying mechanisms and the biological function(s) of endogenous TGFβ are not well understood. To study a possible functional link between autocrine TGFβ and RAC1B, we selected two RAC1B-expressing cell lines, Panc1 (PDAC) and MDA-MB-231 (triple-negative breast cancer), known for their high production and secretion of endogenous TGFβ1 [98,99]. Initially, we observed that RNA interference-mediated knock-down of the *TGFB1* gene was associated with a decrease in basal levels of phospho-p38 and prevented the phosphorylation/activation of p38 by rhTGFβ1 (which is of the delayed type, peaking at 2 h) without affecting total levels of p38 [74]. This was indicative of a requirement of *TGFB1* for efficient activation of p38 by exogenous TGFβ1. We then observed that the synthesis and secretion of autocrine TGFβ1 was stimulated by RAC1B [100] (Figure 3), whereas the related RAC1 decreased the levels of endogenous TGFβ1 in culture supernatants, providing another example of the reciprocal control of a potentially oncogenic response by RAC1B and RAC1. Moreover, the knock-out or knock-down of either *RAC1B* [13] or *TGFB1* [100] resulted in downregulation of *SMAD3* but not vice versa. From this we concluded that RAC1B promotes SMAD3 expression through intermittent induction of *TGFB1* (Figure 3). The realization that *TGFB1* induces SMAD3 now provided a mechanistic link to the failure of cells to activate p38 (in the absence or presence of exogenous TGFβ) when *TGFB1* is knocked down. Since we had shown earlier that p38 activation by TGFβ1 involves intermittent SMAD3-dependent transcriptional activation of *GADD45**β* [101], the decrease in SMAD3 protein following *TGFB1* knock-down likely prevents GADD45β expression and hence p38 activation. The ability of autocrine TGFβ1 to induce SMAD3 protein expression, together with the finding that non-C-terminally phosphorylated SMAD3 can inhibit invasion (see above) led us to propose that the endogenous *TGFB1* gene is involved in mediating the anti-EMT and anti-migratory effect of RAC1B. Intriguingly, knock-down of *TGFB1* or antibody-mediated neutralization of autocrine TGFβ1 in the culture supernatant enhanced rather than decreased both the expression of *SNAI1* or *SNAI2* and the cells’ migratory activity [100]. Since RAC1B, autocrine TGFβ1, and SMAD3 all inhibited migration, and co-depletion of either *RAC1B* and *TGFB1* or *RAC1B* and *SMAD3* failed to provide an additional or synergistic effect over those with depletion of only one gene, we proposed the RAC1B-autocrine TGFβ-SMAD3 axis to represent a novel tumor suppressor pathway [100].

### 4.4. RAC1B Favors the Expression of Extracellular TGFβ Inhibitors

Biglycan (BGN), a member of the family of small leucine-rich secreted proteoglycans, is induced by TGFβ1 in a SMAD3- [57], SMAD4- [102] and p38 MAPK [48,95]-dependent manner and has been identified as a potent inhibitor of EMT and cell invasion in vitro in both murine and human PDAC cells [74,103]. We have shown that RAC1B enhances basal and TGFβ1-induced upregulation of BGN through its ability to induce SMAD3 [13]. The ability of RAC1B to induce BGN mRNA expression in the absence of exogenous TGFβ was retained with a C-terminal serine phosphorylation-resistant mutant of SMAD3 [13], indicating a TGFβ-independent control of BGN. A similar scenario was shown for SMAD3-mediated upregulation of E-cadherin in gastric cancer cells, which involves transcriptional induction of microRNA-200 (miR-200) and subsequent repression of ZEB1 [87]. Members of the miR-200 family form an autoregulatory feedback loop with ZEB1 or ZEB2, which acts as a critical switch point in phenotype conversion during EMT, with miR-200 promoting epithelial differentiation and ZEB1/2 promoting mesenchymal differentiation [104]. This suggests the exciting possibility that the targeting of BGN by the RAC1B-autocrine TGFβ-SMAD3 pathway introduced above also involves the miR-200-ZEB1 autocrine loop. Due to its ability to bind and sequester TGFβ in the extracellular matrix and to prevent its binding to the signal transducing receptors, BGN may be able to protect nearby tumor cells from chronic paracrine (over)stimulation by stromal TGFβ.

### 4.5. Other Possible but Still Hypothetical Targets

Cells have a remarkable ability to rapidly induce receptor transport to the cell surface, thus enhancing the availability of receptors for TGFβ binding and hence the cells’ TGFβ responsiveness [26]. This can occur in response to TGFβ ligand, which thus amplifies its own response [26,35], and this mechanism of response amplification depends on AKT activation and the ALK5 kinase [26,35]. Although the impact of RAC1B on AKT has not yet been analyzed, the ability of RAC1B to downregulate both ALK5 and PAR2, the latter of which might act as a chaperone in anterograde vesicular transport of ALK5, suggests that RAC1B can dampen this ligand-induced response amplification. Furthermore, although a direct role of RAC1 or RAC1B in the intracellular trafficking of TGFβ receptors has not yet been described, it is nevertheless conceivable that RAC1B interferes with TGFβ receptor internalization or endocytotic trafficking, given the involvement of RAC1 in endosomal transport via interaction with Rab proteins including Rab5a [34] and the functional antagonism between RAC1B and RAC1. If so, this will have immediate consequences for signaling activity and possibly on the balance of Smad vs non-Smad signaling. Pursuing this further, it is tempting to speculate that RAC1B favors clathrin-mediated endocytosis (which promotes TGFβ-induced Smad signaling), while blocking membrane raft/caveolae-mediated endocytosis (which facilitates TGFβ-induced non-Smad signaling). Another issue worth following, particularly in cancer cells, is the question of how RAC1B-driven autocrine TGFβ production affects receptor internalization and trafficking and if chronic exposure of the cells to autocrine TGFβ can change their response towards paracrine (stromal) TGFβ.

It is well documented that a portion of both RAC1 [105] and RAC1B [23] is localized to the nucleus; however, very little is known regarding the function and biological significance of nuclear RAC1 or RAC1B. In the nucleus, both isoforms can interact with a very different set of proteins and may, therefore, display novel functional activities. For instance, nuclear RAC1 has been shown to promote cell division [106] and its nucleocytoplasmic shuttling drives nuclear shape changes and tumor invasion [107]. Since TGFβ is a master regulator of both processes, assuming an interaction of RAC1 or RAC1B with Smad transcriptional complexes, or the intracellular domain of ALK5 (TβRI-ICD) that can act as a transcription factor to activate the *SNAI1* and *MMP2* genes [108] (Figure 2), in the nucleus is not too far-fetched. Downregulation of ALK5 by RAC1B will likely also decrease the availability of TβR1-ICD and consequently reduce its DNA-binding activity, consistent with the inhibitory effect of RAC1B on *SNAI1* and *MMP2* expression [74].

## 5. How Can the Differential Interactions of RAC1B with TGFβ Signaling Be Integrated with the Proposed Role of RAC1B as a Tumor Suppressor?

Our observations on RAC1B-TGFβ signaling cross-talk have been mostly made in cancer cells with a well-to-moderately differentiated phenotype and a poor-to-moderately invasive potential, because only these express sufficient amounts of RAC1B. Consistent with this, we found RAC1B to be abundantly expressed in the benign pancreatic ductal epithelial cell line, HPDE [12], suggesting that high RAC1B expression is a general feature of normal epithelial cells lining the pancreatic ducts. In contrast, in cells exhibiting a mesenchymal, highly invasive phenotype, RAC1B levels were low or undetectable. Very recent data from our laboratory suggest that the loss of RAC1B expression is a direct consequence of chronic exposure to TGFβ and is associated with the acquisition of a stem-like phenotype [85]. Collectively, these observations prompted us to postulate for RAC1B a role as gatekeeper of the epithelial phenotype, an inhibitor of mesenchymal transdifferentiation as well as a genuine tumor suppressor with respect to TGFβ1-driven oncogenesis [12] (Figure 4). However, at first glance, the proposed role as a tumor suppressor was not readily compatible with the effect of RAC1B on some individual components of the TGFβ signaling pathway. For instance, the RAC1B-driven USP26-SMAD7-mediated degradation of ALK5 and the autocrine TGFβ1-SMAD3-BGN-mediated sequestration of exogenous, i.e., stromal cell-derived TGFβ is supposed to decrease TGFβ signaling, eventually resulting in downregulation of tumor-promoting MAPK/ERK and RAC1 signaling (Figure 4, left-hand side). However, lower ALK5 levels will inevitably also impair TGFβ—but not necessarily BMP-induced tumor-suppressive Smad2/3 signaling. The surprising discovery of RAC1B being able to induce both SMAD3 [13] and SMAD4 [85] protein expression can be interpreted as an attempt to compensate for the loss of Smad activation by activated ALK5 to ensure execution of tumor-suppressive functions such as growth arrest and apoptosis, despite low levels of ALK5 and/or exogenous TGFβ that reaches the receptors on the cell surface (Figure 4, left-hand side). Indeed, it has been shown that even low-level Smad signaling is able to provide a tumor-protective effect. In addition, RAC1B-driven upregulation of SMAD3 and SMAD4 helps to maintain a differentiated epithelial phenotype by promoting the expression of E-cadherin [87,109,110], inhibiting EMT [12] and cell motility, and even stimulating MET [111] (Figure 4, left-hand side). These pro-Smad signaling effects are probably enhanced by RAC1B’s ability to inhibit activation of ERK2, thereby reducing ERK-mediated cross-inhibition of Smad signaling (see Section 4.2.2.). In contrast, in cells with low RAC1B expression, the above0-described regulatory interactions are reversed, eventually resulting in the preponderance of pro-tumorigenic outcomes as a result of high levels of ALK5 and/or greater access of TGFβ ligand to the receptors (Figure 4, right-hand side). The tumor suppressor function of RAC1B thus relies on selective promotion of Smad over non-Smad signaling and hence may be directed selectively towards TGFβ signaling. This is consistent with data from some genetically modified mouse models, in which RAC1B overexpression collaborated with other oncogenic pathways to promote carcinogenesis [18]. In conclusion, RAC1B might serve both as a tumor suppressor and as an oncogene, depending on the genetic and cellular context and/or the stage of neoplastic progression.

## 6. Concluding Remarks

Our data on RAC1B and TGFβ signaling that focused on classical and cancer-relevant cellular responses clearly point to RAC1B being a potent negative regulator of TGFβ-dependent EMT, cell motility, and growth inhibition. To accomplish this, RAC1B appears to function largely, but not solely, as an antagonist of RAC1. This became even more evident when we studied in detail the effect of RAC1B on the regulation of central components of the TGFβ signal transduction pathway regardless of whether these act in a positive (i.e., ALK5) or negative (i.e., SMAD7, USP26) fashion. Although RAC1 is known to promote mesenchymal differentiation in line with its (only) positive interactions with TGFβ, RAC1B seems to maintain a differentiated epithelial phenotype and prevent the (RAC1-driven) EMT process [12]. Since control of cellular differentiation states or the transition between them involves a plethora of genetic and epigenetic events, there are many genes and epigenetic mechanisms to discover that are targeted by RAC1B, and we predict that many of these will be controlled by RAC1 and RAC1B in a reciprocal fashion. Hence, the final outcome of a particular response in a given tissue or cell type will likely depend on the RAC1B:RAC1 ratio, as speculated earlier [9]. Given the potent inhibition of TGFβ-induced changes in morphology, motility and epithelial/mesenchymal marker expression, RAC1B is likely to also impact other EMT-associated responses, such as cancer stem cell generation and apoptosis sensitivity/chemoresistance. Moreover, RAC1B may also target other type I receptors involved in relaying the TGFβ signal or inducing EMT, i.e., ALK2 [29], ALK1 [26], and other members of the TGFβ superfamily of ligands that utilize ALK5, such as growth differentiation factor-11 (GDF11) or myostatin, or even type I receptors for activins and Nodal (ALK4, ALK7) or BMPs (ALK3, ALK6).

During the process of studying the cross-talk of RAC1B with TGFβ signaling we realized that this small GTPase can serve as a tool to probe and better understand the mechanistic details of TGFβ signal transduction. Moreover, our strong confidence in RAC1B being a tumor suppressor led us to hypothesize that its newly identified targets should themselves be anti-oncogenic. Rigorously sticking to this concept allowed us to verify a hitherto unappreciated anti-migratory function for SMAD3 [13] and autocrine TGFβ1 [100], two factors that had before been considered classical tumor promoters. Although the majority of our observations were made in selected adenocarcinoma cell lines of pancreatic and breast origin, we surmise that these mechanisms also operate in other but not necessarily all epithelial cells or cancer types. Rather, the specific mechanisms may depend on differentiation state or may be heterogeneous among the cancer cells of a tumor as a result of its clonal origin/evolution [113].

Finally, the close association of RAC1B expression with a differentiated epithelial phenotype and tumor-suppressive Smad signaling on the one hand and its low or absent expression in undifferentiated mesenchymal-type tumor cells with tumor-promoting non-Smad, i.e., ERK signaling on the other hand corresponds well to the function of TGFβ as a tumor suppressor in early-stage cancers and a tumor promoter in their late-stage counterparts, the shift of which is known as the “TGFβ paradox” [114,115]. It is therefore tempting to speculate that only after the levels of RAC1B or the RAC1B:RAC1 ratios in the cancer cells have dropped to a certain threshold are TGFβ’s oncogenic activities freed from inhibition to allow the further development and metastatic progression of human carcinomas.

## Figures and Tables

**Figure 1 cancers-12-03475-f001:**
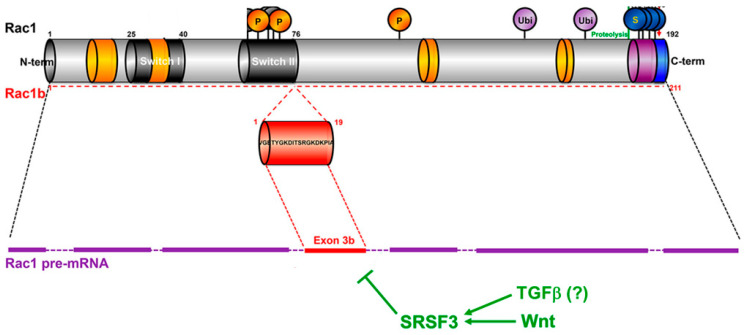
Primary structure of RAC1 and RAC1B and regulation of *RAC1* splicing. The sites of posttranslational modifications are indicated by circles. P, phosphorylation; Ubi, ubiquitination; S, SUMOylation. For details see the text. Adapted from [3], modified.

**Figure 2 cancers-12-03475-f002:**
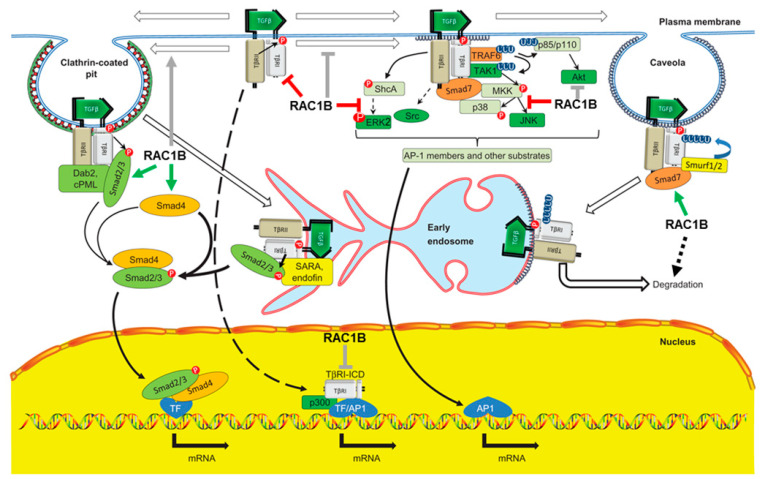
Schematic illustration of signaling via TGFβ receptors initiated at different intracellular locations and sites of inhibition by RAC1B. TGFβ binding to its receptors on the cell surface activates both Smad-mediated and non-Smad signal transduction pathways. Receptors are distributed between different compartments/microdomains of the plasma membrane. In the clathrin-coated pits (left-hand side) activated TGFβ receptors bind and phosphorylate SMAD2/3, initiating the canonical Smad pathway. In the lipid rafts/caveolae of the plasma membrane (right-hand side) the receptors preferentially associate with SMAD7. This inhibitory Smad negatively regulates Smad-mediated signaling by competing with SMAD2/3 for interaction with ALK5/TβRI and recruits the E3 ubiquitin-protein ligases Smurf1 and Smurf2, which direct ubiquitin-dependent degradation of the TGFβ receptor complex (right-hand side). SMAD7 can also function as an adaptor protein, facilitating the activation of non-Smad signaling pathways (center). In early endosomes (EEs) Smad-mediated signaling is further enhanced by accessory proteins (SARA and endofin). In the nucleus, the intracellular domain of ALK5 (TβRI-ICD) can act as a transcription factor to regulate the expression of genes like *SNAI1* and *MMP2*. Stimulatory interactions of RAC1B are denoted by green arrows, inhibitory ones by red lines. Gray arrows or lines indicate the still hypothetical nature of the interactions. Adapted from [31], modified.

**Figure 3 cancers-12-03475-f003:**
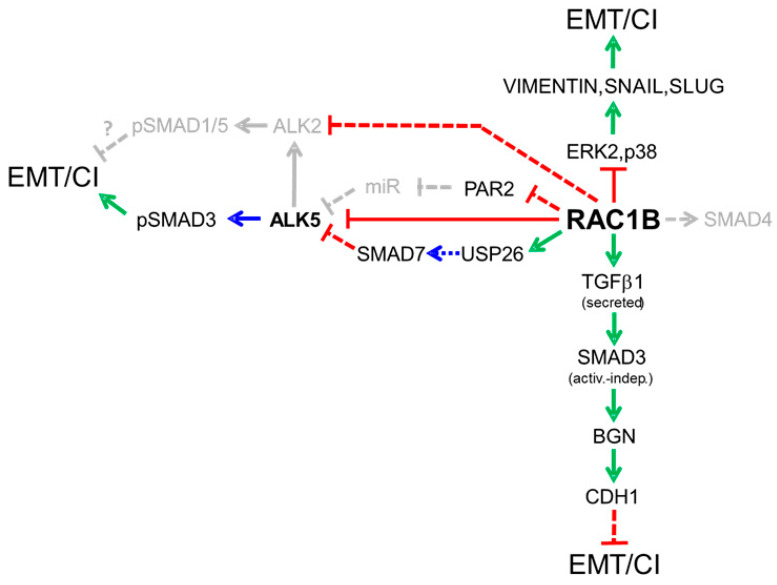
Diagram summarizing our previous findings on the regulatory interactions of RAC1B with TGFβ signaling mediators and MAPKs that in a TGFβ-dependent or independent manner inhibit epithelial–mesenchymal transition (EMT) or cell invasion (CI) in pancreatic epithelial cells. Green arrows denote stimulation of expression, blue arrows stimulatory post-translational modifications and red lines inhibition of expression. Stippled lines indicate the possible involvement of intermediary factors. Pathways in gray color are either hypothetical or have not yet been published.

**Figure 4 cancers-12-03475-f004:**
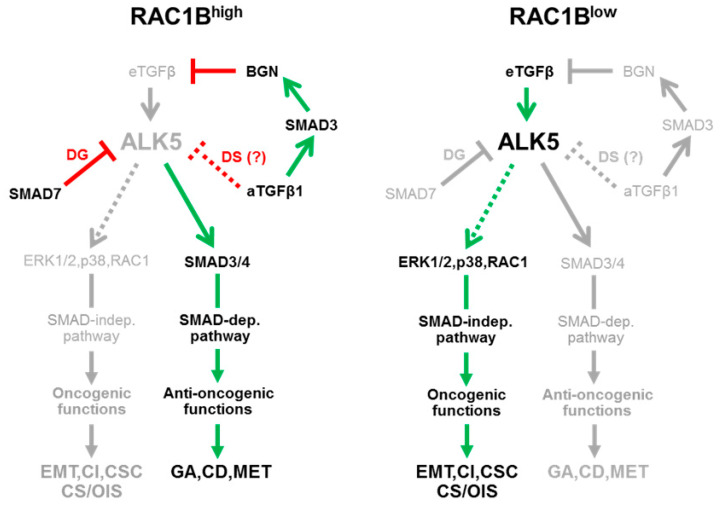
Schematic diagram illustrating the proposed role of RAC1B in regulating TGFβ signaling in pancreatic tumor cells. Left-hand side, RAC1B inhibits receptor activation via (i) induction of SMAD7 and SMAD7-mediated degradation (DG) of ALK5, (ii) autocrine TGFβ (aTGFβ)-mediated desensitization (DS) of cells towards exogenous TGFβ, and (iii) aTGFβ-SMAD3-mediated induction of BGN, which sequesters exogenous TGFβ (eTGFβ) in the pericellular space and prevents it from binding to the receptors. As a result, the activities of the Smad3/4 and non-Smad (ERK1/2, p38, RAC1) signaling pathways are repressed, but the decrease in Smad signaling is partially rescued by RAC1B-driven upregulation of SMAD3 and SMAD4 (not shown). This constellation suppresses oncogenic programs like EMT, cell invasion (CI), and cancer stem cell (CSC) formation, but allowing tumor-suppressive functions such as growth arrest (GA), cell death/apoptosis (CD), and mesenchymal–epithelial transition (MET) to proceed. Right-hand side, under conditions of low or absent RAC1B expression, ALK5 expression is increased and subsequent activation of non-canonical signaling predominates over Smad signaling, an effect that is enhanced by the inability of the cell to produce additional SMAD3 and SMAD4 proteins. Non-canonical TGFβ signaling pathways, in particular ERK and RAC1, can now promote their oncogenic functions. Cellular senescence (CS) and oncogene-induced senescence (OIS), although driven by MEK-ERK signaling [92,93,94], are generally considered tumor-suppressive mechanisms but may also be tumor-promoting in some instances [112]. Stimulatory interactions are indicated by arrows and inhibitory interactions by lines (green/red = activated, gray-shaded = inactivated). Stippled lines denote still hypothetical interactions (aTGFβ) or the possibility of ALK5-independent activation (ERK1/2, RAC1).

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
