# Peer review of "The Small GTPase RAC1B: A Potent Negative Regulator of-and Useful Tool to Study-TGFβ Signaling"

_cancers, 2020, doi:10.3390/cancers12113475_

Round 1
Reviewer 1 Report
This manuscript discusses a new role of RAC1B, an understudied RAC1 isoform, on regulating TGFb signaling pathway.
Based on recent studies of RAC1B in model pancreatic cancer cell lines, the authors propose that RAC1B has an inhibitory role for TGFb signaling pathway at multiple molecular levels.
This manuscript will provide deeper insights into TGFb signaling pathway as well as the understudied, but important RAC1B role in cancer biology.
Author Response
Dear Editor, dear Carel:
This letter of submission is accompanied by our revised manuscript entitled:
“The Small GTPase RAC1B: A Potent Negative Regulator of - and Useful Tool to Study - TGFb Signaling”
We are indebted to the reviewers for their valuable comments and suggestions and have done our best to incorporate these into the revised version of our manuscript (highlighted in the “track changes” mode). In particular, we have included a new figure (Figure 4) to be used also as graphical abstract.
We believe that the reviewers’ critiques have substantially improved the quality of our manuscript and we are looking forward to its final acceptance in Cancers.
Sincerely yours,
Hendrik Ungefroren
Reviewer 2 Report
The review by Ungefroren et al., is clear and interesting. Only minor issues were identified:
- Line 76 introduces DPC4 but at that stage it is out of context.
- The paragraph beginning line 95 should be clearer about what "expression" means in terms of the mRNA vs the protein for Rac1B.
- Line 226 states that Rab5 is required for Rac1 activation, but the citation for that is relatively weak (#34) and this statement seems too definitive.
- The sentence beginning on line 318 is a run-on. It also begins "The resembles..", which is likely a typo for "This resembles...".
- Line 410 "antagonize" should be "antagonizes".
- Line 614 "which" should be "that".
- One thing to consider is the use of i.e. and e.g.. It might be the style of the journal, but conventionally these are followed by a comma.
Author Response
Dear Editor, dear Carel:
This letter of submission is accompanied by our revised manuscript entitled:
“The Small GTPase RAC1B: A Potent Negative Regulator of - and Useful Tool to Study - TGFb Signaling”
We are indebted to the reviewers for their valuable comments and suggestions and have done our best to incorporate these into the revised version of our manuscript (highlighted in the “track changes” mode). In particular, we have included a new figure (Figure 4) to be used also as graphical abstract.
We believe that the reviewers’ critiques have substantially improved the quality of our manuscript and we are looking forward to its final acceptance in Cancers.
Sincerely yours,
Hendrik Ungefroren
The review by Ungefroren et al., is clear and interesting. Only minor issues were identified:
1) Line 76 introduces DPC4 but at that stage it is out of context.
Response: We agree with this comment and have deleted the second part of this sentence referring to DPC4.
2) The paragraph beginning line 95 should be clearer about what "expression" means in terms of the mRNA vs the protein for Rac1B.
Response: As requested, this has been specified at three locations in this paragraph.
3) Line 226 states that Rab5 is required for Rac1 activation, but the citation for that is relatively weak (#34) and this statement seems too definitive.
Response: We agree with the reviewer and have weakened the statement accordingly.
4) The sentence beginning on line 318 is a run-on. It also begins "The resembles..", which is likely a typo for "This resembles...".
Response: This typo has been corrected.
5) Line 410 "antagonize" should be "antagonizes".
Response: This typo has been corrected.
6) Line 614 "which" should be "that".
Response: Terms have been changed, as requested.
7) One thing to consider is the use of i.e. and e.g.. It might be the style of the journal, but conventionally these are followed by a comma.
Response: As requested, commata have been added behind these abbreviations.
Reviewer 3 Report
The Review article of Pr Hendrik Ungefroren et al aims to summarize the current state of understanding of the implication of the RAC1B -a splice variant of the RAC1 small GTPase- as a negative regulator of TGF-beta signaling in the context of carcinogenesis.
The manuscript provides an informative and didactic introduction of the RHO GTPases, and more especially RAC1 and RAC1B splice variant, followed by a very detailed description of TGF-beta signaling and its deregulation in cancer cells. After presenting these features, the Authors analyze exhaustively the interplay of RAC1/RAC1B at the different levels of TGF-beta signaling, including the paracrine/ autocrine factor, its receptors, the downstream canonical and non-canonical downstream signaling pathways, and the biological impact in term of proliferation, migration and senescence. The manuscript clearly displays the different facets of TGF-beta signaling and how RAC1B selectively antagonizes or promotes the related biological effects.
This Review article is well written and constitutes an up-to-date analysis of the literature concerning RAC1/ TGF-beta signaling pathways, their cross-talk and their significance during carcinogenesis. This review provides therefore a suitable contribution for "Cancers".
Minor points
The quality of the Figures should be improved. The Authors may request the original files of Figures 1 & 2 and adapt them in the context of the Review.
The Authors might try to propose a graphical abstract/ scheme depicting TGF-beta/ TGF-beta receptors, their downstream signaling pathways (including canonical Smad2/3, and non- canonical RAC1 pathways) and their biological effects (proliferation, invasiveness, senescence), combined with the crosstalk/ inhibitory impact of RAC1B. I am aware that this is not an easy task, due to the multiple pathways and biological effects controlled by TGF-beta, the involvement of feedback loops, and the different levels of regulation of these pathways by RAC1B.
Some references quote should be corrected:
Lines 98-100: references 9 & 10 have been switched
Line 322: reference 10 may not be adequate
Line 524: reference 10 may be replaced by reference 9 and/or 73
The Review arouses many interesting questions that will require further experimental investigations (see below). Some of them might be further discussed in the manuscript.
Is the high levels of RAC1B in differentiated cancer cell lines and the poor expression in mesenchymal or basal-like cell lines a cause or a consequence? How do these observations compare to corresponding tumor and control tissues? In some subtypes of cancers RAC1B is ectopically expressed, being almost undetectable in control tissue. How to conciliate these observations with a role of RAC1B in the maintenance of cell differentiation? Is RAC1B expressed under physiological conditions in a restricted subset of normal (stem) cells that can be prone to transformation? Is RAC1B transiently and ectopically expressed at early stage of carcinogenesis to overcome OIS and promote cell proliferation, and its downregulation subsequently required for further neoplastic progression?
In some genetically modified mouse models, RAC1B overexpression cooperates with other oncogenic pathways to promote carcinogenesis. Does it mean that RAC1B suppressor activity is directed towards selective signaling pathways, e.g. TGF-beta signaling? depends on the genetic and cellular context, and/or the stage of the neoplastic progression? Thus, RAC1B might serve both as a tumor suppressor and as an oncogene?
Author Response
Dear Editor, dear Carel:
This letter of submission is accompanied by our revised manuscript entitled:
“The Small GTPase RAC1B: A Potent Negative Regulator of - and Useful Tool to Study - TGFb Signaling”
We are indebted to the reviewers for their valuable comments and suggestions and have done our best to incorporate these into the revised version of our manuscript (highlighted in the “track changes” mode). In particular, we have included a new figure (Figure 4) to be used also as graphical abstract.
We believe that the reviewers’ critiques have substantially improved the quality of our manuscript and we are looking forward to its final acceptance in Cancers.
Sincerely yours,
Hendrik Ungefroren
1) The quality of the Figures should be improved. The Authors may request the original files of Figures 1 & 2 and adapt them in the context of the Review.
Response: We thank the reviewer for this suggestion. We have, indeed, attempted to request the original files but in one case it was difficult to contact the authors by Email and in the other case we did not receive a response from the publisher at all despite several inquiries via Email. For this reason and in order to avoid a further delay in publishing our article we prefer to leave the figures 1 and 2 as they stand.
2) The Authors might try to propose a graphical abstract/ scheme depicting TGF-beta/ TGF-beta receptors, their downstream signaling pathways (including canonical Smad2/3, and non- canonical RAC1 pathways) and their biological effects (proliferation, invasiveness, senescence), combined with the crosstalk/ inhibitory impact of RAC1B. I am aware that this is not an easy task, due to the multiple pathways and biological effects controlled by TGF-beta, the involvement of feedback loops, and the different levels of regulation of these pathways by RAC1B.
Response: We thank the reviewer for this suggestion and have included another figure (new Figure 4 to be used also as graphical abstract) that encompasses as many interactions as possible. However, for the sake of clarity, it was not possible to include all pathways and feedback loops in this figure.
Some references quote should be corrected:
Lines 98-100: references 9 & 10 have been switched
Line 322: reference 10 may not be adequate
Line 524: reference 10 may be replaced by reference 9 and/or 73
Response: We apologize for the confusion. All reference numbers have been corrected.
3) The Review arouses many interesting questions that will require further experimental investigations (see below). Some of them might be further discussed in the manuscript.
Is the high levels of RAC1B in differentiated cancer cell lines and the poor expression in mesenchymal or basal-like cell lines a cause or a consequence? How do these observations compare to corresponding tumor and control tissues? In some subtypes of cancers RAC1B is ectopically expressed, being almost undetectable in control tissue. How to conciliate these observations with a role of RAC1B in the maintenance of cell differentiation? Is RAC1B expressed under physiological conditions in a restricted subset of normal (stem) cells that can be prone to transformation? Is RAC1B transiently and ectopically expressed at early stage of carcinogenesis to overcome OIS and promote cell proliferation, and its downregulation subsequently required for further neoplastic progression?
In some genetically modified mouse models, RAC1B overexpression cooperates with other oncogenic pathways to promote carcinogenesis. Does it mean that RAC1B suppressor activity is directed towards selective signaling pathways, e.g. TGF-beta signaling? depends on the genetic and cellular context, and/or the stage of the neoplastic progression? Thus, RAC1B might serve both as a tumor suppressor and as an oncogene?
Response: As recommended by the reviewer, we have discussed several of these interesting questions/issues in section 5.